# Impacts of Increasing $CO_2$ on Diurnal Migrating Tide in the Equatorial Lower Thermosphere

**Masaru Kogure[1], In-Sun Song[1], Huixin Liu[2], and Han-Li Liu[3]**

[1]Department of Atmospheric Sciences, Yonsei University, Seoul, South Korea.

[2]Department of Earth and Planetary Science, Kyushu University, Fukuoka, Japan

[3]High Altitude Observatory, National Center for Atmospheric Research, Boulder, CO, USA

*Corresponding author:* Masaru Kogure (masarukogure@yonsei.ac.kr)

**Abstract.** We investigate the impacts of increased $CO_2$ concentration on migrating diurnal tide (DW1). A future climate simulation is conducted using a WACCM-X model, with surface $CO_2$ levels increasing according to the RCP 8.5 scenario. The DW1 (1,1) mode, a propagating tide peaking near the equator, exhibits a statistically significant positive trend in a range of 20–70 km, and a significant negative trend in a range of 90–110 km. The positive trend is likely driven by a reduction in atmospheric density in the mesosphere and enhanced equatorial convective activity, while the negative trend appears in the mesosphere, which overwhelms the positive trend. Two potential mechanisms may explain the negative trend. First, increasing $CO_2$ enhances mesospheric stability, reducing tidal vertical wavelengths. In our simulation, equatorial temperatures around ~50–70 km become cooler than those in ~70–90 km. This strong cooling could be linked to $CO_2$ mixing and transport, as well as the contraction of the mesospheric ozone layer due to atmospheric descent induced by $CO_2$-driven cooling. Second, stronger convective activity intensifies gravity wave generation, increasing gravity wave diffusion in the mesosphere. This strong convective activity also likely intensifies the tide below ~70 km. While our positive DW1 trend is consistent with McLandress and Fomichev (2006), the negative trend in the lower thermosphere contrasts with their results. This discrepancy might arise because their model used a time-independent diffusion coefficient, whereas WACCM-X accounts for $CO_2$-driven changes in gravity wave diffusion. The negative trend is confirmed in SABER observation for the last two decades, while the positive trend is not verified.

## 1. Introduction

Anthropogenic $CO_2$ emissions from fossil fuel combustion and industrial processes have risen steadily since the Industrial Revolution and are projected to continue increasing in the future (IPCC, 2023). This rise has significantly elevated atmospheric $CO_2$ concentrations, contributing to global warming in the troposphere. Global warming has altered geographical precipitation patterns, including their frequency and intensity, as well as atmospheric circulation patterns (e.g., Arias et al., 2021; Chou & Neelin, 2004; Feng et al., 2019; Lau et al., 2013). On the other hand, atmospheric layers above the tropopause, such as the stratosphere, mesosphere, thermosphere, and ionosphere, have been cooling as $CO_2$ concentration has increased, and their altitudes have descended (e.g., Akmaev and Fomichev, 1998; Arias et al., 2021; Cnossen, 2020; Emmert et al., 2010; Emmert, Fejer, et al., 2004; Emmert, Picone, et al., 2004; Garcia, 2021; Garcia et al., 2019; Jonsson et al., 2004; Keating et al., 2000; Kogure et al., 2022; Laštovička, 2021; Laštovička et al., 2008, 2012; H. Liu et al., 2020, 2021; Marcos et al., 2005; Ogawa et al., 2014; Qian et al., 2011; Ramesh & Sridharan, 2018; Roble & Dickinson, 1989; Zhang et al., 2011). Akmaev and Fomichev (1998) demonstrated that cooling/contraction of the atmosphere can lead to apparent warming at specific altitudes in the lower thermosphere (~100–120 km) due to the large temperature gradient therein. Similarly, the ionosphere also descends in response to $CO_2$ cooling, altering the vertical profiles of ion and electron density (see Figure 16.5 in Laštovička, 2021).

In addition to changes in temperature, the dynamical effects of increasing $CO_2$ on the thermosphere have been demonstrated by Liu et al. (2020), showing for the first time an enhanced circulation and significant changes in thermospheric tidal activities. Since thermal tides play a key role in thermosphere–ionosphere dynamics, understanding the impact of rising $CO_2$ levels on these tides is essential for predicting thermospheric climate change. Tides are classified into two categories based on their dynamical characteristics: trapped and propagating tides (Chapman & Lindzen, 1970). Trapped tides from in-situ forcing dominate in the middle and upper thermosphere (above ~140 km altitude), whereas tides in the lower thermosphere primarily originate in the troposphere and stratosphere and propagate upward (Yamazaki et al., 2014; Yamazaki and Siddiqui, 2024). Using the whole atmosphere model Ground-to-topside Atmosphere Ionosphere model for Aeronomy (GAIA), Liu et al. (2020) found an enhancement of the migrating tides (DW1) below 200 km, and a reduction of the migrating semidiurnal tides

(SW2) throughout most of the thermosphere. Ma et al. (2025) confirmed similar tidal trends in a long-term future projection simulation of WACCM-X (The Whole Atmosphere Community Climate Model with thermosphere and ionosphere extension). Regarding the lower thermosphere, McLandress and Fomichev (2006) examined the impacts of doubled $CO_2$ on propagating DW1 tides, which primarily occur in the equatorial MLT (mesosphere–lower thermosphere) region. Using a linear tidal model, they compared three scenarios: (1) present-day $CO_2$ levels, (2)

doubled $CO_2$ with present-day sea surface temperatures, and (3) doubled $CO_2$ with sea surface temperatures adjusted accordingly. These scenarios were based on present-day observations and simulations conducted with the Canadian Middle Atmosphere Model (CMAM). Their results suggested that doubling $CO_2$ increases tropospheric water vapor and its radiation of solar heating, amplifying DW1 tides in the MLT by ~1 K (10–15%). However, their linear tidal model did not incorporate changes in gravity wave (GW) diffusion, despite its significant role in tidal dissipation,

because it used a time-independent eddy diffusion coefficient, which is independent of GW drag (McLandress, 2002). Additionally, since propagating DW1 tides are generated by tropospheric disturbances and modulated by stratospheric and mesospheric background conditions, uncertainties in lower atmospheric states directly affect predictions of future DW1 variability.

       This study investigates the impact of increasing $CO_2$ on propagating DW1 tides using the state-of-the-art

model, WACCM-X. We used a 2° horizontal resolution version of WACCM-X. Although this model cannot resolve most gravity waves, it includes both orographic and non-orographic gravity wave parameterizations.

       Section 2 describes the specifications of WACCM-X, the gravity wave parameterizations, simulation setup, and our analysis method, specifically the Hough mode decomposition. Section 3 presents the results, showing a positive trend in propagating DW1 tides in the stratosphere and a negative trend in the lower thermosphere. Section

4 discusses potential mechanisms for enhanced tidal dissipation in the MLT region. Section 6 shows the tidal trends observed by SABER (Sounding of the Atmosphere using Broadband Emission Radiometry) between 2002 and 2024. Finally, Section 6 summarizes our findings, compares them with those of McLandress and Fomichev (2006), and discusses the limitations of this study. Unless otherwise stated, "tides" hereafter refers to propagating DW1 tides.

## 2.   Data and Analysis

### 2.1. WACCM-X Simulation

       This study uses the same long-term simulation as those used in Ma et al. (2025) and Pedatella et al. (2025). Briefly, it is a 90-year simulation (2000–2089) using CESM/WACCM-X version 2.2 with coupled ocean. The model has latitudinal and longitudinal resolutions of 1.9° × 2.5°, respectively. Its vertical resolution decreases with

altitude, transitioning from 0.16 density scale height at 100 hPa to 0.25 density scale height at 1 hPa. Above 1 hPa, the resolution is fixed at 0.25 density scale height (e.g., ~1.6, ~1.4, and ~2.6 km at ~70, ~90, and ~110 km altitudes, respectively). The model time step is 15 minutes, and we used monthly mean output parameters for our analysis. WACCM-X simulates the entire atmosphere, from Earth's surface to the thermosphere and ionosphere, with a model top at ~500-700 km altitude. It includes orographic and non-orographic GW parameterizations, accounting for three

GW sources: deep convection (Alexander et al., 2021; Beres et al., 2005), frontogenesis (Richter et al., 2010), and winds over complex terrain (Garcia et al., 2017; Scinocca and McFarlane, 2000). WACCM-X also includes a cumulus convection parameterization (Zhang and McFarlane, 1995), which is coupled with the parameterization for convectively-generated GWs (Beres et al., 2005). The model also solves chemical reactions for five ion species, electrons, and 74 neutral species, including ozone. A detailed description of the dynamical processes in WACCM-X

(version 2) is provided by H. L. Liu et al. (2018).

In this simulation, the horizontally uniform $CO_2$ concentration is specified near the surface and follows historical observations up to 2014, after which it follows the Representative Concentration Pathway 8.5 (RCP 8.5) scenario. Above the surface, $CO_2$ concentrations vary due to atmospheric transportation and eddy diffusion. Although the simulation runs until the end of 2089, we analyze only the data before July 2069, as the surface $CO_2$ concentration exceeds the upper limit of the Fomichev non-LTE $CO_2$ cooling scheme (720 ppm). In equatorial regions, the maximum error in the non-LTE $CO_2$ cooling rate near the mesopause is ~1 K day$^{-1}$ at 360 ppm and ~2 K day$^{-1}$ at 720 ppm (Fomichev et al., 1998). This error seems linearly increasing from 150 to 720 ppm. Figure 1a shows the global mean $CO_2$ concentrations, smoothed over one-year, at the surface and altitudes of 50 km and 100 km, respectively. Solar cycle activity follows the Coupled Model Intercomparison Project (CMIP6) (O'Neill et al., 2016); that is, it is specified using historical observations up to 2014, and from 2015 onward, it is simulated using the solar forcing based on historical observational data from 1850 to 1924 levels (Figure 1b). Since future solar forcing cannot be reliably predicted, we used past values following CMIP6.

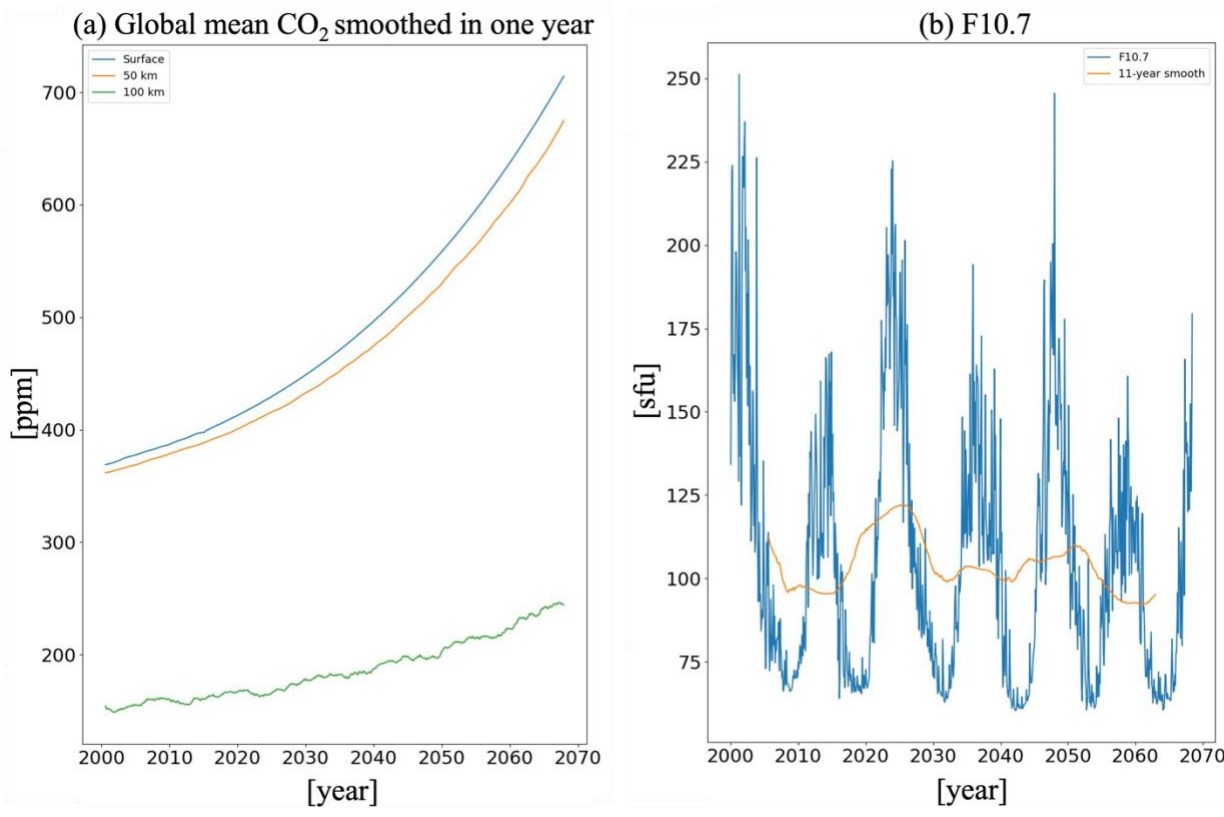

Figure 1. (a) Global mean $CO_2$ concentrations, smoothed over one year. The blue, orange, and green lines represent $CO_2$ concentrations at the surface, 50 km, and 100 km altitudes, respectively. (b) F10.7 solar flux (blue) and its 11-year smoothed trend (orange).

## 2.2. DW1 (1, 1) Mode

Monthly mean temperature perturbations of the DW1 tide were derived from the output monthly mean diurnal components (the parameter names are "Temperature 24hr. cos and sin coeff.") using a discrete Fourier transform. These perturbations were then convolved with the (1,1) Hough mode function at each pressure level to derive the (1,1) mode amplitudes. These amplitudes were subsequently interpolated to geometric height for each month. Unless otherwise stated, other parameters are also processed in the same manner. The Hough mode functions are solutions of Laplace's tidal equation under the assumptions of an isothermal atmosphere with no background wind. The (1,1) Hough mode is predominantly generated by solar heating of tropospheric water vapor and propagates upward into the lower thermosphere. This mode is dominant in the equatorial region, accounting for over ~90% of DW1 amplitudes, as its energy is concentrated within 30°N/S, with peak temperature amplitudes at the equator. The Hough function was calculated using the normalized Associated Legendre Polynomial (ALP)

expansion method (Groves, 1981), implemented through a Python module developed in this study. This module is based on the MATLAB program developed by Wang et al. (2016), but it also includes functionality for calculating the Hough modes of D0 tidal waves, which cannot be computed in the original MATLAB program.

### 3. **Results**

Figure 2a shows the amplitudes of the (1,1) Hough mode in the 70−110 km geometric height range, representing the MLT region. The output geometric height ($z^*$) is defined in the WACCM-X simulation as:

$$z^* = \frac{R_e}{R_e - z_{GP}} z_{GP}, \tag{1}$$

where $R_e$ and $z_{GP}$ are the Earth's radius (6371 km) and geopotential height, respectively (Neale et al., 2010). The amplitudes were smoothed using a 2-year boxcar window because the stratospheric quasi-biennial oscillations (QBO) significantly modulate tidal wave amplitudes (Hagan et al., 1999; Liu et al, 2017; Kogure and Liu, 2021; Xu et al., 2009). It should be noted that the WACCM-X configuration used in this study does not internally generate a QBO due to its resolution (H. Liu et al., 2018). Our simulation is imposed by relaxing the equatorial zonal winds to observations for the period of 2000–2015. Beyond 2015, it is imposed to observations from 1959 to 2015. Because this study focuses on multi-decadal trends, the QBO phase is unlikely to affect the inferred long-term trend. The amplitudes peak between 100 and 110 km, and these peak altitudes remain constant until 2069, while the peak amplitudes gradually decrease over time. For example, maximum amplitudes were 16–18 K during 2001–2010 but decreased to 14–16 K during 2058–2067, representing a reduction of approximately 2 K (~10%). This negative trend can also be seen on pressure coordinates (Figure 2b). Because of the facilitation of comparison with observations, this study focuses on the results of geometric height coordinates. Figures 2c–g present time series of the monthly mean tidal amplitudes averaged over the 20–30 km, 30–50 km, 50–70 km, 70–90 km, and 90–110 km altitude ranges (green crosses). The 2-year smoothed amplitudes are shown in black, while their linear trends are depicted in red. Table 1 summarizes the slopes and their standard errors for the linear fits. Here, the standard error denotes the one-sigma (~68%) confidence interval of the regression, which is calculated by the SciPy module (Virtanen et al., 2020). Notably, the amplitudes in the 90–110 km range significantly decrease over time, while those below 70 km significantly increase. In the 70 and 90 km range, the magnitude of the slope is 4–5 times smaller than its standard error, indicating no significant trend. This result suggests that tidal source activity in the troposphere (solar heating due to water vapor absorption and latent heating) intensifies, leading to strong DW1 tides below ~70 km. Conversely, above ~70 km, tidal dissipation increases, offsetting the positive effects of enhanced source activity and reducing the amplitude above ~ 90 km. It should be noted that atmospheric long-term trends can be time-dependent (Lastovicka and Jelínek, 2019). Although our multi-decadal analysis shows statistically significant tidal trends in each layer, except for 70–90 km, this does not imply a monotonic change at the rates listed in Table 2 over short periods (e.g., a single decade). To mitigate the influence of the tidal time-dependent variability and interannual fluctuations, we estimate trends using the full 69-year term, which is ~6 times longer than the 11-year solar cycle.

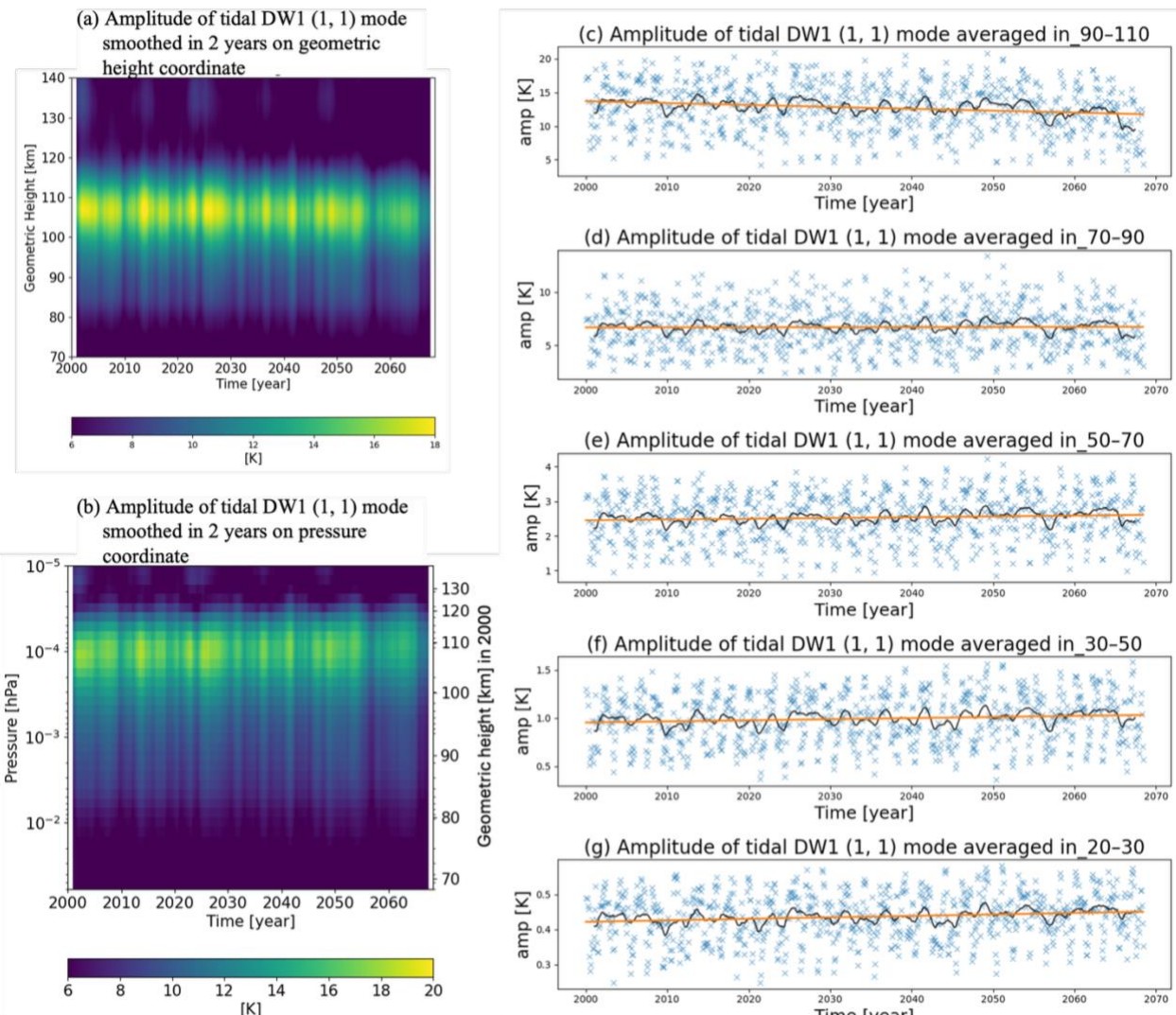

Figure 2. (a) Amplitudes of the (1,1) Hough mode smoothed over two years on geometric height (a) and pressure coordinates (b), respectively. The right-axis in (b) shows the geometric height in 2000 corresponding to the left-axis (pressure) for reference. Note that geometric height sinks with atmospheric cooling. (c–g) Time series of monthly mean amplitudes averaged over 20–30 km, 30–50 km, 50–70 km, 70–90 km, and 90–110 km. The green crosses represent monthly values; the black lines denote 2-year smoothed values; the red lines represent linear trends.

| Height [km] | Trend of the amplitude [K/year] | Standard error of trend [K/year] |
|---|---|---|
| 20-30 | $4.2 \times 10^{-4}$ | $1.2 \times 10^{-4}$ |
| 30-50 | $1.1 \times 10^{-3}$ | $4.6 \times 10^{-4}$ |
| 50-70 | $2.3 \times 10^{-3}$ | $1.2 \times 10^{-4}$ |
| 70-90 | $8.0 \times 10^{-4}$ | $3.7 \times 10^{-3}$ |
| 90-110 | $-2.8 \times 10^{-2}$ | $6.3 \times 10^{-3}$ |

Table 1. Slopes and their standard errors of the linear fit shown in Figure 2(c-g).

To further investigate the impact of increasing $CO_2$ on the tidal vertical propagation, particularly above ~70 km, we compared the tidal amplitudes during two periods: January 2003–December 2013 and December 2050–

November 2061. These periods were selected because they have similar mean F10.7 values and standard deviations (95.9±26.6 sfu in 2003–2013 vs. 95.8 ±23.4 sfu in 2050–2061). However, the mean $CO_2$ concentration at the surface in 2050–2061 is 608 ppm, ~58% higher than in 2003–2013. Figure 3a illustrates the ratio of the amplitudes between 2003–2013 and 2050-2061. Across most of the 40-82 km range, amplitude in 2050-2061 is significantly larger (by up to ~9%) than that in 2003-2013. However, above ~72 km, the ratio gradually decreases with altitude and drops below 100% at ~82 km, indicating that amplitudes in 2050-2061 are smaller than those in 2003-2013 above this altitude. These differences in the tidal amplitudes are directionally consistent with the linear trends at each layer in the full term (69 years); thus, they provide a qualitative indication of the multi-decadal trend, while the precise magnitudes should be interpreted with caution.

The ratio shown in Figure 3a depends on variations in tidal source activity in the troposphere, atmospheric density, and propagation conditions. The vertical structure of wave amplitude, $A_{(z)}$, can be approximately described based on Forbes and Vicent (1989):

$$A_{(z)} = A_{(z_0)} \sqrt{\frac{\rho_{(z_0)}}{\rho_{(z)}}} \exp \int_{z_0}^{z} -\sigma_{i_{(z)}} \, dz. \tag{2}$$

Here, $z$ and $z_0$ represent altitude and the altitude of the wave source (i.e., the troposphere for the DW1 tide), respectively. $\rho_{(z)}$ denotes atmospheric density, and $\sigma_{i_{(z)}}$ is the damping factor due to dissipation. Using equation (1), the ratio of tidal amplitude at $z$ between 2003–2013 and 2050–2061 can be expressed as:

$$\frac{A_{2050-2061_{(z)}}}{A_{2003-2013_{(z)}}} = \frac{A_{2050-2061_{(z_0)}}}{A_{2003-2013_{(z_0)}}} \sqrt{\frac{\rho_{2050-2061_{(z_0)}}}{\rho_{2003-2013_{(z_0)}}}} \sqrt{\frac{\rho_{2003-2013_{(z)}}}{\rho_{2050-2061_{(z)}}}} \exp \int_{z_0}^{z} -(\sigma_{i_{2050-2061_{(z)}}} - \sigma_{i_{2003-2013_{(z)}}}) \, dz. \tag{3}$$

In the WACCM-X simulation, the average value of $\sqrt{\frac{\rho_{2050-2061_{(z_0)}}}{\rho_{2003-2013_{(z_0)}}}}$ in 0–17 km is 100.2 $\pm$ 0.6%, which is almost unity. Thus, equation (2) simplifies to:

$$\frac{A_{2050-2061_{(z)}}}{A_{2003-2013_{(z)}}} = \frac{A_{2050-2061_{(z_0)}}}{A_{2003-2013_{(z_0)}}} \sqrt{\frac{\rho_{2003-2013_{(z)}}}{\rho_{2050-2061_{(z)}}}} \exp \int_{z_0}^{z} -(\sigma_{i_{2050-2061_{(z)}}} - \sigma_{i_{2003-2013_{(z)}}}) \, dz. \tag{4}$$

Therefore, the amplitude ratio shown in Figure 3a can be explained by changes in the tidal amplitude at $z_0$ (source activity), the atmospheric density, and tidal damping factor due to dissipation (propagation condition). Figure 3b shows the square root of the atmospheric density ratio between 2003–2013 and 2050–2061, $\sqrt{\frac{\rho_{2003-2013_{(z)}}}{\rho_{2050-2061_{(z)}}}}$. This ratio gradually increases with altitude above 50 km, reaching ~4% at ~72 km. A similar increasing trend is observed in the amplitude ratio shown in Figure 3a. These results suggest that the tidal positive trend in 50–70 km is driven not only by enhanced source activity but also by the decrease in atmospheric density in the mesosphere. However, above ~72 km, the amplitude ratio decreases with altitude, even though $\frac{\rho_{2003-2013_{(z)}}}{\rho_{2050-2061_{(z)}}}$ continues to rise. These vertical variations suggest that tidal damping factor ($\sigma_{i_{2050-2061_{(z)}}}$) increases significantly above ~72 km in 2050-2061. This enhanced damping counteracts the tidal amplification caused by the increased source activity and reduced density, leading to a decline in tidal amplitude near ~82 km. As a result, a negative tidal trend is projected in the MLT region.

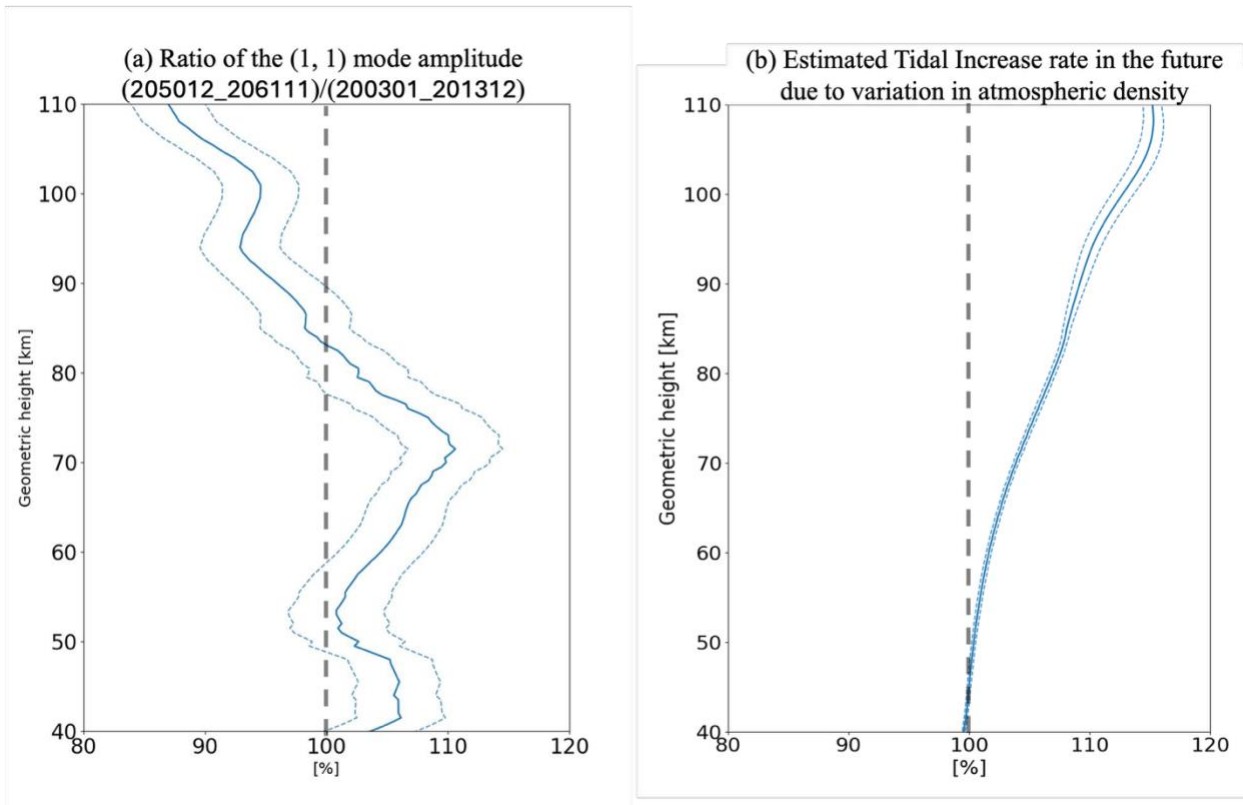

Figure 3. (a) Percentage ratio of the tidal amplitude, averaged over 2050-2061 and normalized by the average in 2003–2013. (b) Estimated future tidal increase rate due to changes in atmospheric density ($\frac{\rho_{2003-2013}(z)}{\rho_{2050-2061}(z)}$). The dashed lines denote the mean values plus/minus their standard errors.

4.    **Impact of Increasing CO₂ Concentration on Tidal Dissipation**

The previous section demonstrated that tidal amplitudes in the MLT region decrease over time (i.e., with increasing $CO_2$ concentration), even though they increase below ~82 km. This decrease in tidal amplitudes is likely attributed to enhanced dissipation above ~72 km. This section explores the potential mechanisms responsible for the enhanced dissipation. To the best of our knowledge, tidal damping in the middle atmosphere is modulated by three primary factors: vertical wavenumber (Forbes and Vincent, 1989; Kogure and Liu, 2021), gravity wave breaking (Forbes and Vincent, 1989; Lu and Fritts, 1993; Meyer, 1999), and meridional shear of zonal wind around 18° in latitude, where tidal zonal wind perturbations peak (Kogure and Liu, 2021; Mayr and Mengel, 2005; McLandress, 2002). This section focuses on changes in vertical wavenumber and gravity wave diffusion, comparing conditions between 2003–2013 and 2050-2061. Discussion of the shear is omitted because the magnitudes of the zonal mean zonal wind shear were found to have decreased in 2050-2061 compared to 2003–2013 (see Figure S1 in the supplement), which is favorable for the DW1 tide and reduces tidal damping.

**4.1. Vertical wavenumbers in 2003–2013 vs. 2050–2061.**

Forbes and Vincent (1989) demonstrated that tidal dissipation due to eddy diffusion is approximately proportional to the square of the vertical wavenumber (i.e., inversely proportional to the square of the vertical wavelength). Here, we derived the vertical wavenumbers of the (1,1) mode from the phases shown in Figure S2 (in supplement), using the least-squares method applied over five vertical steps (~9 km) with a step size of ~1.4 km, following the approach of Kogure and Liu (2021). Figure 4a compares the vertical wavenumbers for the periods 2003–2013 (blue) and 2050–2061 (orange). Between ~55 and ~82 km, the vertical wavelengths in 2050–2061 are shorter than those in 2003–2013. Figure 4b shows the difference in vertical wavenumber values between the two periods. A positive difference exceeding one standard error appears throughout most of the ~54–82 km range, with a peak at ~74 km altitude. These results suggest that shorter vertical wavelengths (larger vertical wavenumbers) in 2050–2061 contribute to stronger tidal dissipation and reduced tidal amplitudes. However, in 2050–2061, the tidal amplitude is larger within the altitude range of the shorter vertical wavelengths (~54–82 km; compare Fig. 3a and Fig. 4b). This apparent inconsistency indicates that the dissipation due to the decreases in vertical wavelengths is hidden by the enhancement of the tide due to its source activity and reduction of the atmospheric density. It further suggests that, above ~82 km, parameterized gravity wave drag is the primary contributor to the amplitude reduction. The effect of gravity waves will be discussed in Section 4.2.

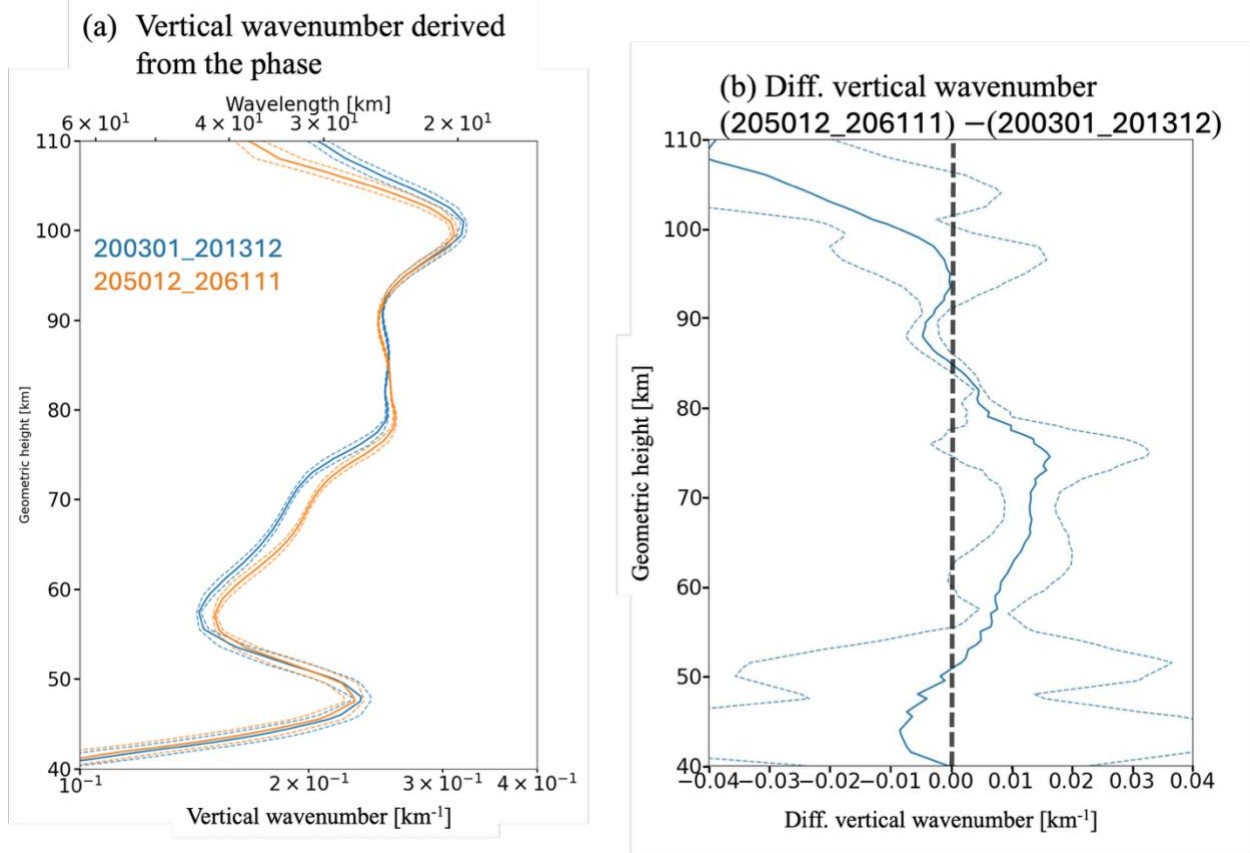


Figure 4. (a) Vertical wavenumbers of DW1 (1,1) mode tides derived from tidal phase values (shown in Figure S2). The upper x-axis denotes vertical wavenumber on a logarithmic scale, and the lower x-axis denotes the corresponding vertical wavelength. The orange and blue lines show vertical wavenumbers averaged over January 2003–December 2013 and December 2050–November 2061, respectively. (b) Difference in vertical wavenumbers

between 2003–2013 and 2050–2061. Dashed lines denote the standard errors.

Under the WKB approximation and assuming no gravity wave drag, the local tidal vertical wavenumber at a given colatitude, $k_{z(z,\theta)}$, is described as (Forbes and Vincent, 1989; Kogure and Liu, 2021):

$$k_{z(z,\theta)}^2 = \frac{N_{(z,\theta)}^2}{gh'_{(z,\theta)}} - \frac{1}{4H_{(z,\theta)}^2}, \tag{5}$$

where $N, g, H,$ and $\theta$ represent the buoyancy frequency, gravitational acceleration, scale height, and colatitude, respectively. $h'$ is the Doppler-shifted equivalent depth, expressed as:

$$h'_{(z,\theta)} = h\left(1 + \frac{\mathrm{u}_{(z,\theta)}}{C_0 \sin\theta}\right)^4, \tag{6}$$

where $h, C_0,$ and $u$ are the equivalent depth (0.69 km for the (1,1) mode) under no background wind, the magnitude of the migrating diurnal tide phase speed at the equator (~465 ms$^{-1}$), and the background zonal wind, respectively.

Figure 5a shows the difference in the local vertical wavenumber ($k_{z(z,\theta)_{2050-2061}} - k_{z(z,\theta)_{2003-2013}}$) calculated using Eq. (5). Below ~92 km, the differences in local wavenumbers at latitudes below ~20°N/S qualitatively match those derived from tidal phase analysis (Figure 5b), particularly the statistically significant positive peak within the ~60–80 km range. Since $k_{z(z,\theta)}$ depends on $N, H,$ and $h'$, changes in vertical wavenumber $\delta k_{z(z,\theta)}$ due to variations in these parameters can be approximated by a Taylor series expansion:

$$\delta k_{z(z,\theta)} \approx \frac{N\delta N}{kgh'} - \frac{N^2 \delta h'}{2kgh'^2} + \frac{\delta H}{4kH^3} + O(\delta^2). \tag{8}$$

The first and second terms represent the effects of changes in buoyancy frequency and Doppler-shifted equivalent depth (i.e., zonal mean zonal wind), respectively. The third and fourth terms account for changes in the scale height

and high-order terms, which are relatively small and can be neglected (not shown). Figure 5b, c illustrate the effects of buoyancy frequency and Doppler-shifted equivalent depth, respectively. The buoyancy frequency effect agrees

with the characteristics of $\delta k_{z(z,\theta)}$ below ~82 km, exhibiting a statistically significant positive peak at ~72 km and a significant negative peak near ~48. Notably, the difference between $\delta k_{z(z,\theta)}$, and the buoyancy frequency effect at ~72 km over the equatorial region is less than ~$5 \times 10^{-4}$ km$^{-1}$. Conversely, within the positive $\delta k_{z(z,\theta)}$ region (~60–80 km altitude), the magnitude of the Doppler-shifted effect is almost insignificant and approximately three times smaller than that of the buoyancy frequency effect within 20°N/S, although around the negative $\delta k_{z(z,\theta)}$ layer

( ~80–90 km), the magnitude of the Doppler-shifted effect is significant in the equatorial region and larger than that of the buoyancy frequency. Therefore, the increase in buoyancy frequency likely shortens the vertical wavelengths between ~60 and ~80 km altitudes, thereby enhancing tidal dissipation in the mesosphere.

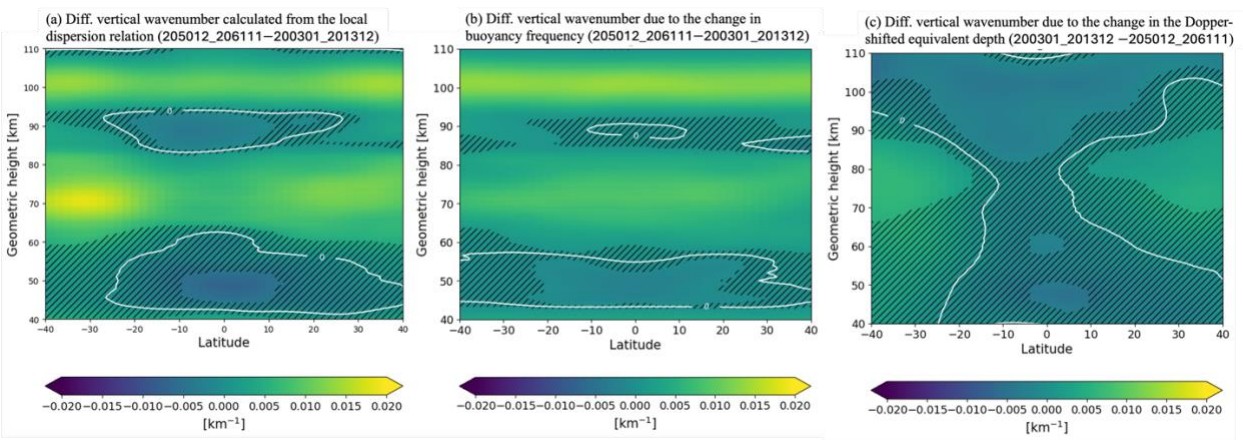

Figure 5. (a) Difference in local vertical wavenumber computed from Eq. (5). (b) Effects of buoyancy frequency changes due to increasing CO$_2$ on the local vertical wavenumber. (c) Same as (b), but showing Doppler-shifted effects. Hatched areas indicate regions where the differences are statistically insignificant (i.e., within 1-sigma standard error).

To further explore the relationship between the increasing CO$_2$ cooling and the increased buoyancy frequency, we compare temperatures and their vertical gradients between 2003–2013 and 2050–2061. Figure 6 illustrates the differences in zonal mean temperatures (6a) and their vertical gradients (6b), respectively. The negative temperature difference in ~50–70 km is significantly larger (~-9 K at maximum) than that (~-2 K) in ~75–85 km. This strong cooling in ~50–70 km increases atmospheric stability, thereby strengthening buoyancy frequency

in ~62–82 km.

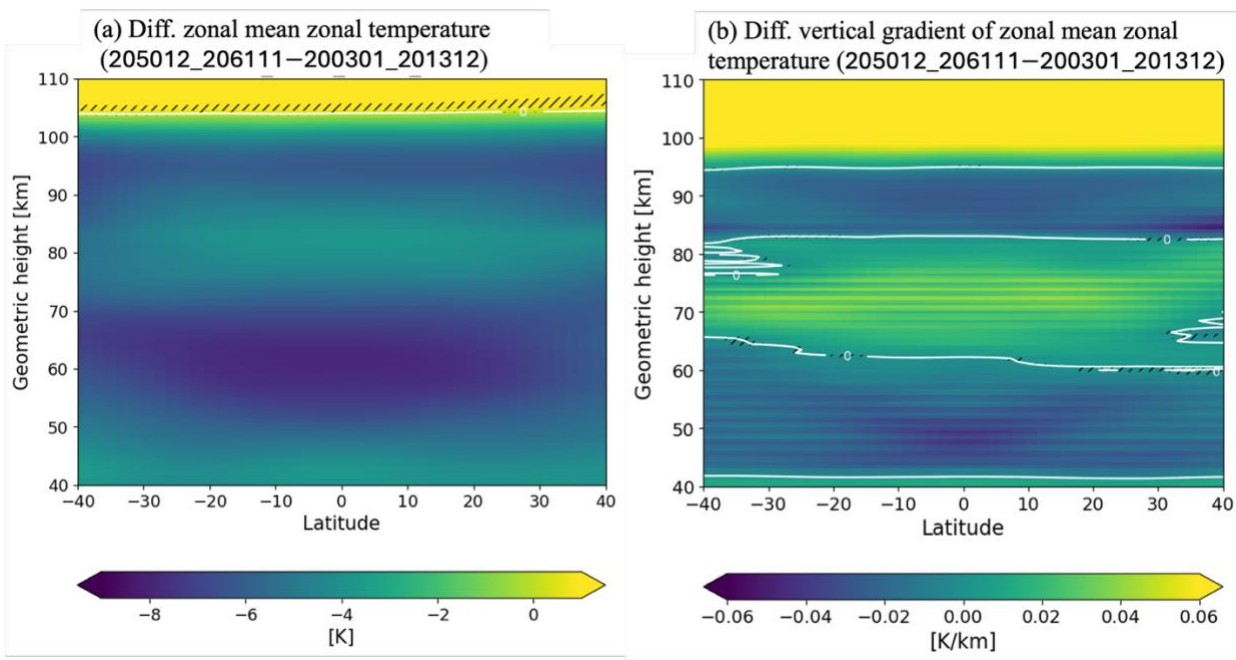

Figure 6. (a) Difference in zonal mean temperature between 2050–2061 and 2003–2013. (b) Same as (a), but showing differences in vertical temperature gradients. Hatched area denotes regions where the differences are statistically insignificant (i.e., within 1-sigma standard error).

Next, we examine the mechanism responsible for the cooler temperatures in the lower to middle mesosphere (~50–70 km). Temperatures in this layer are influenced not only by $CO_2$ cooling but also by $O_3$ heating via ultraviolet absorption (Garcia, 2021; Garcia et al., 2019; Jonsson et al., 2004; Lübken et al., 2013). Figure 7 shows the differences in $CO_2$ and $O_3$ concentrations, calculated by subtracting the mean values in 2003–2013 from those in 2050–2061. $CO_2$ concentration significantly increases uniformly by ~200 ppm in ~40–80 km, with the rate of increase sharply declining above ~ 80 km. This indicates that $CO_2$ is well mixed up to ~80 km, which results in stronger cooling below that altitude. $O_3$ concentration significantly decreases by up to ~0.05 ppm in ~53-79 km and increases by up to ~0.16 ppm in ~78-92 km during 2050–2061, contributing to cooler temperatures in the lower mesosphere and warmer temperatures in the upper mesosphere. This vertical change in $O_3$ concentration is likely due to atmospheric descent caused by the $CO_2$ cooling, as both the positive and negative peaks of the $O_3$ concentration descend by a few kilometers in the future (shown later in Fig. 8c).

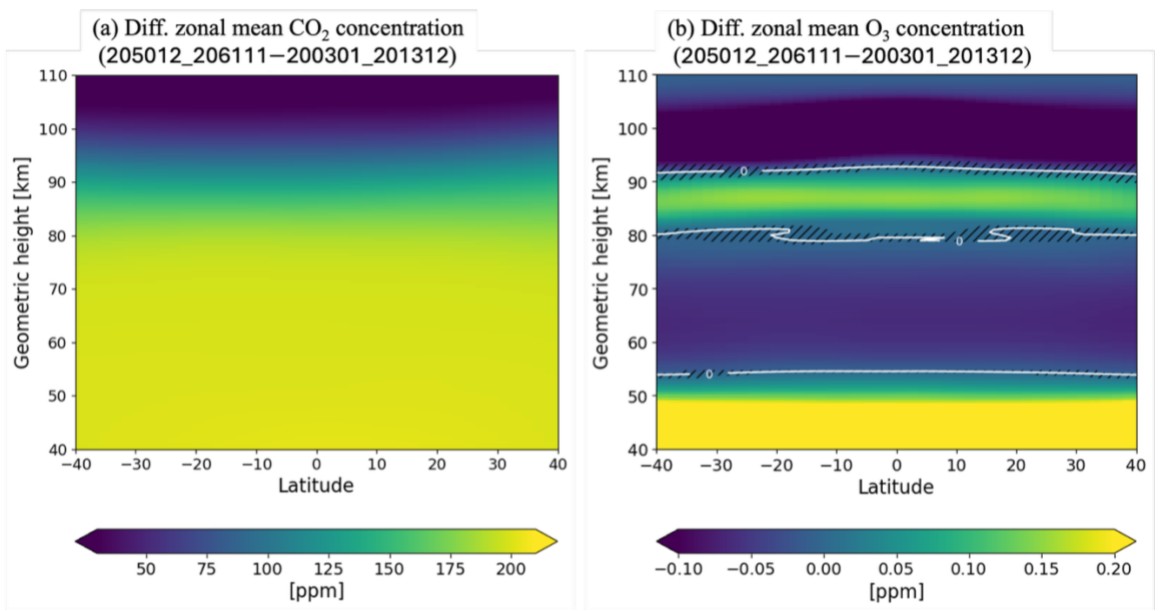

Figure 7. (a) Relative difference in $CO_2$ concentration. (b) Same as (a), but for the $O_3$ concentration. Hatched areas indicate regions where the differences are statistically insignificant (i.e., within 1-sigma standard error).

Figure 8 shows vertical profiles of temperature (8a), $CO_2$(8b), and $O_3$ concentrations(8c) at 1°N, along with their differences (2050–2061 minus 2003–2013) in panels (8d)–(8f). $CO_2$ concentrations in Figure 8(b) remain nearly constant up to ~80km and then decrease sharply above that, suggesting that $CO_2$ concentrations are mixed well up to ~80 km. This vertical feature is seen in its difference (Figure 7a) as aforementioned. These results support the idea that $CO_2$ cooling is more effective below ~80 km due to the high concentration. Regarding $O_3$ concentrations in Figure 8(c), the altitudes of the local minimum (~78 km) and maximum (~93 km) shift downward in 2050–2061. This contraction leads to increased $O_3$ above and decreased $O_3$ below the ~78 km altitude, resulting in a local maximum (~+0.15 ppm at ~86 km) and minimum (-0.05 ppm at 62 km) in Figure 8f. These levels correspond to a local minimum (~-3 K at ~85 km) and maximum (~-8 K at ~62 km) in temperature decrease between 50 and 90 km in Figure 8d. This correspondence supports that the mesospheric ozone vertical variation contributes to the strengthened stability in the lower and middle mesosphere.

To summarize this subsection, $CO_2$ likely induces cooling throughout ~40–80 km than ~80–110 km due to its mixing and transport, while $O_3$ contributes to cooling in ~53–79 km and warming in ~79–92 km due to the downward shift of the $O_3$ layer. This downward shift results in a pronounced cold region around ~60 km and intensifies atmospheric stability in ~62–82 km. Thus, the vertical changes in the $O_3$ and $CO_2$ concentrations can qualitatively explain the vertical change in the temperature, although a more quantitative evaluation is needed and is beyond the scope of this study.

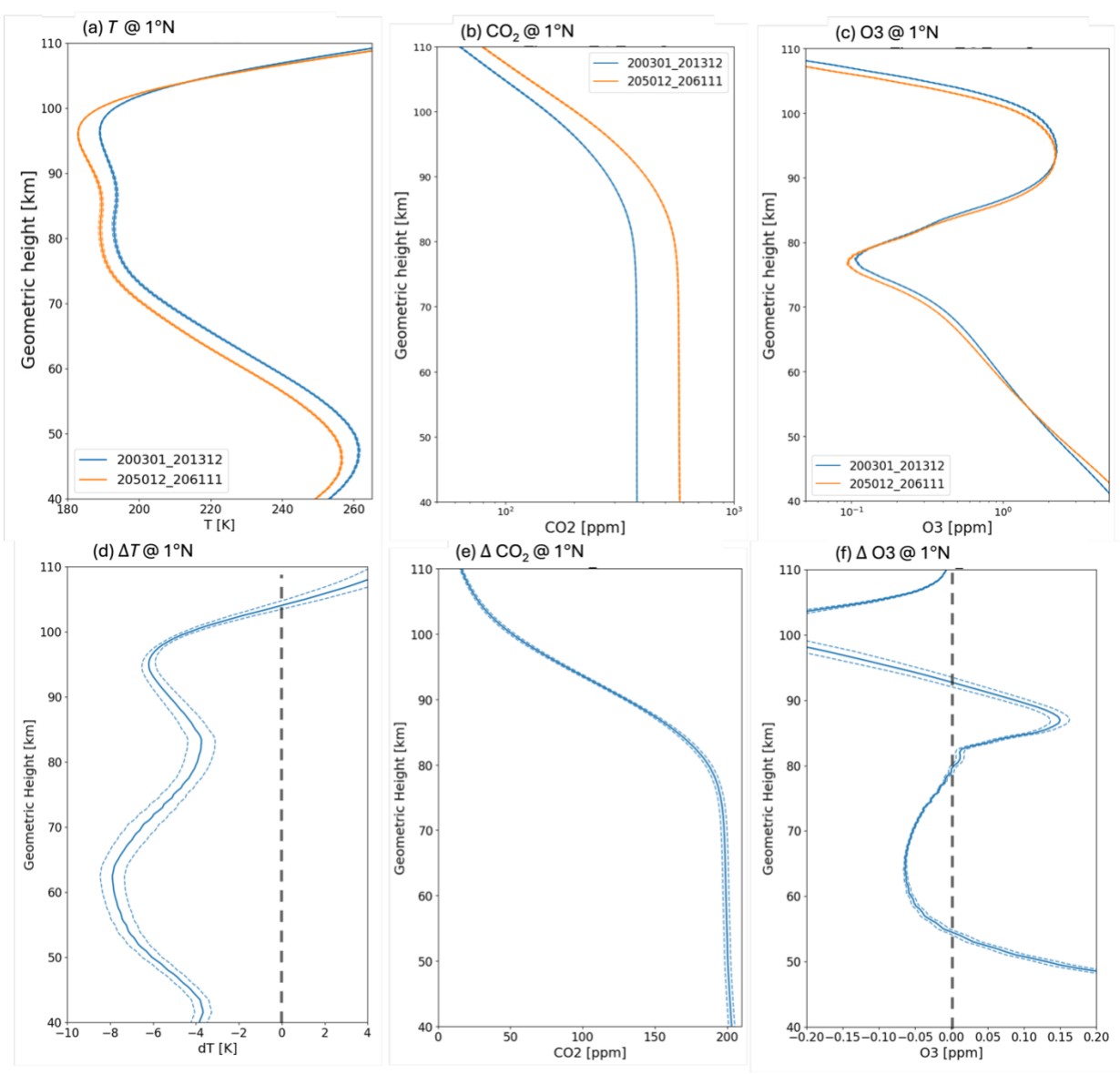

310

Figure 8. (a) Vertical profile of temperature at 1°N in 200301–201312 (blue) and 205012–20611 (orange). (b) same as (a) but for $CO_2$ concentration. (c) same as (a) but for $O_3$ concentration. (d) Difference in temperature at 1°N (205012–20611 minus 200301–201312). (e) same as (d) but for $CO_2$ concentration. (f) same as (d) but for $O_3$ concentration. Dashed lines represent $\pm 1$-sigma standard error intervals.

315

## 4.2. Gravity Wave Drag Parameterization in 2003–2013 vs. 2050–2061.

According to Meyer (1999), gravity wave breaking interacts with tides through two mechanisms: gravity wave diffusion and the diurnal harmonics of gravity wave drags. Here, we focus on gravity wave diffusion, as the diurnal harmonics of the wave drag were not available due to data storage limitations. It should be noted that the wave drag effect depends on a specific gravity wave scheme used (Mayr et al., 1998; McLandress, 1997; Meyer, 1999; Miyahara and Forbes, 1991), whereas gravity wave diffusion consistently damps tides among the schemes (Meyer, 1999). Saturation and breaking of gravity waves induce diffusion, which dissipates atmospheric waves including tides. The diurnal harmonics of the gravity wave drag can amplify the tides in the real atmosphere, depending on their relative phases with the tidal oscillation (Ortland and Alexander, 2006). However, the gravity wave drag based on the Lindzen scheme, which is implemented in WACCM-X, only acts to dissipate the tide (Mayer, 1999). Consistent with this, the tidal amplitude in the MLT decreases in the future run (see Fig.2c and Fig. 3a)

Figure 9 shows the difference in diffusion due to parameterized gravity waves between 2003–2013 and 2050–2061. Between 30°N and 30°S, diffusion significantly increases at nearly all altitudes from 40 km to 110 km, with a pronounced peak around 90 km (~1.3 ms$^{-2}$ at maximum), although a localized decrease appears around 10°N in ~95–100 km. These increases correspond to an ~10% increment from 2003–2013. Since convection is the primary source of gravity waves in equatorial regions, where DW1 tides are concentrated, we examine the difference in the zonal mean precipitation rate. The precipitation rate significantly increases between ~5°S and ~5°N, particularly around ~0°N/S by ~$4.5 \times 10^{-9} \pm$ ~$0.7 \times 10^{-9}$ m $\cdot$ s$^{-1}$, except for its significant decrease at ~10°N by ~$4.1 \times 10^{-9} \pm$ ~$2.2 \times 10^{-9}$ m $\cdot$ s$^{-1}$; the tropical precipitation mostly increases. These changes correspond to the variations in gravity wave diffusion. This increase in tropical precipitation is consistent with findings from previous studies on tropospheric climate change (e.g., Chou and Neelin, 2004; Feng et al., 2019; Lau et al., 2013). In addition, previous numerical modeling studies have reported that increased tropical precipitation intensifies stratospheric GWs and their source activity (Franke et al., 2023; Watanabe et al., 2005). Integrating findings from these previous studies with our simulation, we can lead to the following potential scenario. Increased $CO_2$ concentrations strengthen equatorial convection activity, leading to enhanced tropical gravity wave activity in the stratosphere. These enhanced gravity waves propagate upward and, upon reaching the MLT, saturate and break. This process intensifies diffusion and reduces the tidal amplitudes. The GW diffusion in ~30-40°N is also significantly enhanced, possibly due to increased frontogenesis, which might also contribute to the tidal damping.

It should be noted that the strengthened equatorial convection likely leads to the tidal positive trend below ~70 km with the depression in the atmospheric density in the mesosphere, shown in Figures 2 and 3, as well. However, the tidal dissipation associated with the strengthened stability and GW diffusion in the mesosphere could overwhelm the positive trend, resulting in the significant negative trend in the MLT layer.

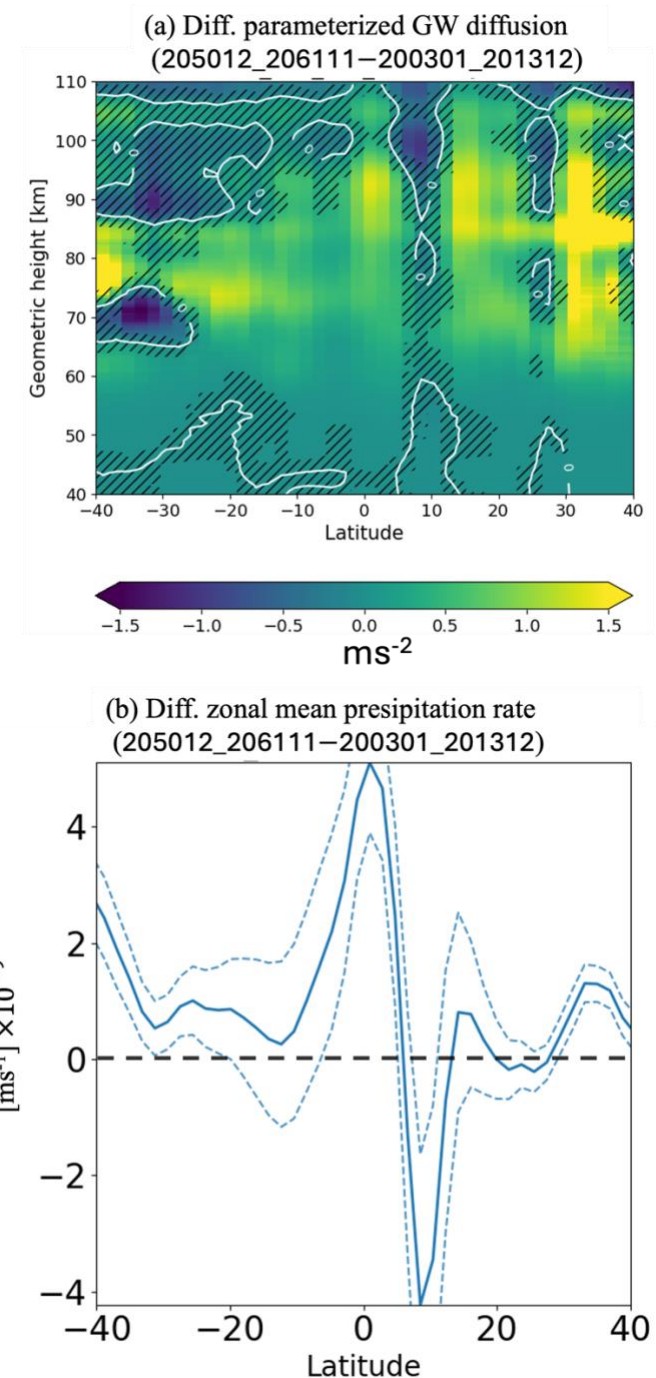

(a) Diff. parameterized GW diffusion
(205012_206111−200301_201312)

(b) Diff. zonal mean presipitation rate
(205012_206111−200301_201312)

Figure 9. (a) Difference in diffusion due to parameterized gravity waves between 2003–2013 and 2050–2061, shown as a function of latitude and geometric height. Hatched areas indicate regions where the differences are statistically insignificant (i.e., within 1-sigma standard error). (b) Difference in zonal mean precipitation rate between 2003-2013 and 2050-2061. Dashed lines represent $\pm$1-sigma standard error intervals.

## 5. DW1 Tidal Trend in SABER in 2002–2024.

To assess observational consistency with the simulation results shown in Section 3, this section examines DW1 tidal trends derived from temperature observations by the SABER instrument aboard NASA's TIMED (Thermosphere Ionosphere Mesosphere Energetics Dynamics) satellite. We analyzed the SABER data spanning 23 years (2002–2024), following the method described by Yamazaki and Siddiqui (2024). The data within the latitude range of 50°N/S are binned into 15° longitude, 5° latitude, 1-hour universal time, and 2 km altitude bins at 3-month intervals. At each latitude and altitude, the binned temperatures are fitted with the following equation:

$$\sum_{s=-4}^{+4} \sum_{\sigma=0}^{3} T_{cos\ s,\sigma_{(\theta,z)}} \cos(\frac{2\pi x}{360°}s - \frac{2\pi t_{UT}}{24}\sigma) + T_{sin_{s,\sigma_{(\theta,z)}}} \sin(\frac{2\pi x}{360°}s - \frac{2\pi t_{UT}}{24}\sigma).$$

Here, $s$ and $\sigma$ represent the zonal wavenumber and frequency, respectively, while $x, \theta, z$, and $t_{UT}$ represent longitude, latitude, altitude, and universal time. The DW1 components (i.e., $T_{cos-1,1}$ and $T_{sin-1,1}$) are decomposed into four Hough mode components (first symmetric and anti-symmetric, propagating and trapped modes) using the least squares fitting method. Since the 23-year term might be insufficiently long than natural seasonal and interseasonal variability, we removed seasonal variability by subtracting the 23-year (2002–2024) averages from each 3-month mean. To further eliminate intraseasonal variability, such as that caused by the El Niño-Southern Oscillation (ENSO) and the Quasi-Biennial Oscillation (QBO), we apply the multiple regression analysis on the deseasonalized values using time, the Oceanic Niño Index (ONI), and zonal wind averaged in 100–10 hPa as independent variables. The ONI data are obtained from the NOAA webpage (https://origin.cpc.ncep.noaa.gov/products/analysis_monitoring/ensostuff/ONI_v5.php), and the zonal wind data are obtained from ERA-5 (Hersbach et al., 2020). We apply the same multiple regression method with the WACCM-X future run in 2002–2024, but the ONI index and the stratospheric zonal wind are obtained from the WACCM-X simulation itself. It should be noted that we apply the singular least squares fitting method to the DW1 amplitudes in SABER, which is the same approach used in the WACCM-X future run in section 3. Both methods yield comparable results, suggesting that the periods of ENSO and QBO are sufficiently short relative to the 23-year term, and their influence on the long-term trend is likely negligible.

| Height [km] | WACCM-X (2002–2024) | SABER (2002–2024) |
| --- | --- | --- |
| 90–110 | $-4.6 \times 10^{-3} \pm 3.9 \times 10^{-2}$ | $-5.1 \times 10^{-2} \pm 3.1 \times 10^{-2}$ |
| 70–90 | $+1.5 \times 10^{-4} \pm 1.7 \times 10^{-2}$ | $+1.3 \times 10^{-2} \pm 3.1 \times 10^{-2}$ |
| 50–70 | $-2.0 \times 10^{-4} \pm 7.2 \times 10^{-3}$ | $+7.6 \times 10^{-3} \pm 9.3 \times 10^{-3}$ |
| 30–50 | $-1.3 \times 10^{-5} \pm 2.5 \times 10^{-3}$ | $-6.6 \times 10^{-3} \pm 7.0 \times 10^{-3}$ |
| 20–30 | $+7.3 \times 10^{-5} \pm 7.3 \times 10^{-4}$ | $+2.3 \times 10^{-3} \pm 3.1 \times 10^{-3}$ |

Table 2. Tidal linear trends in the amplitude of the DW1 (1,1) mode from 2002 to 2024 in WACCM-X future run and SABER.

Table 2 shows the linear trends in the amplitude of the DW1 (1,1) mode from 2002 to 2024 in both WACCM-X and SABER. While WACCM-X in the 23-year period shows no statistically significant trends at any altitudes, SABER reveals a significant negative trend in the 90–110 km region. This result is consistent with the long-term trend (2000–2069) in the WACCM-X future run (see Table 1). No significant trends are seen below 90 km in SABER.

However, we caution that the statistical significance of these results depends on the analysis period. For example, during 2005–2024, the trend of the tides observed by SABER in the 90–110 km range remains negative but is not statistically significant (not shown). In the same manner, the simulated trends in 2002–2024 are inconsistent with those in 2000-2069, implying that the 23-year term is insufficient to drive the $CO_2$ impact in the simulation at least. Continuous observations of the MLT over the next two decades will be essential to obtain more robust evidence.

6. **Conclusion and Discussion.**

   We examined the response of the DW1 tidal (1,1) mode to increasing $CO_2$ concentrations using a long-term WACCM-X simulation spanning from January 2000 to July 2069, following the RCP 8.5 scenario. The analysis reveals two significant responses of the DW1 tide to increasing $CO_2$ levels: (1) Below 70 km altitude, tidal activity increases significantly, likely due to enhanced water vapor and latent heating. Additionally, in 50–70 km, the
depression in the atmospheric density likely contributes to the positive trend. (2) Above 90 km altitude, tidal activity decreases significantly, likely due to increased tidal dissipation above ~70 km. The tidal amplitudes in 90–110 km observed by SABER significantly show a consistent negative trend in 2002–2024, although the trends below 90 km are insignificant. We propose two potential mechanisms contributing to the increased tidal dissipation: a decrease in tidal vertical wavelength and an increase in diffusion due to GW breaking. The decreasing vertical wavelength
contributes to the dissipation in ~52–82 km, peaking at ~75 km, while the increasing GW breaking diffusion contributes in ~45–110 km around the equatorial region, peaking at ~93 km. The shorter vertical wavelength is likely linked to enhanced stability in the mesosphere, as cooling in the lower and middle mesosphere is stronger than in the upper mesosphere. This relatively strong cooling could be attributed to the vertical variations in $CO_2$ and $O_3$ concentrations in the mesosphere. $CO_2$ increases more below ~ 80 km than above, due to its mixing and transport,
which induces stronger cooling throughout ~40–80 km than ~80–110 km. Meanwhile, the mesospheric ozone layer shifts downward, leading to decreased concentrations and cooling within ~53–79 km, and increased concentrations and warming within ~79–92 km. The combination of $CO_2$ and $O_3$ vertical variations intensifies atmospheric stability in ~62–82 km, thereby reducing the tidal vertical wavelengths and amplitudes there. Additionally, the increase in GW diffusion may be attributed to enhanced convective activity, as tropical precipitation is intensified. While this
enhanced convective activity likely strengthens tidal activity and contributes to the tidal positive trend below ~70–80 km, the increased tidal dissipation in the mesosphere overwhelms this positive effect, resulting in the significant negative trend in the MLT layer. Taking into account the clearly increasing negative trend in the future above ~80 km (see Figure 3a), the GW diffusion might contribute to the tidal damping more than the shortened vertical wavelengths.
Our findings for the troposphere and stratosphere agree with those of McLandress and Fomichev (2006), whereas our results for the MLT region show an opposite trend. We believe this inconsistency arises from differences in vertical diffusion. In CMAM, which simulated the background conditions in McLandress and Fomichev (2006), the lower mesosphere became cooler than the upper mesosphere as $CO_2$ concentrations increased, consistent with our simulation results (see Figure 10 in Fomichev et al., 2007). However, unlike WACCM-X, which
accounts for variations in gravity wave diffusion with increasing $CO_2$, the tidal linear model used in McLandress and Fomichev (2006) employed a time-independent vertical diffusion coefficient. This likely contributes to the differences in outcomes.
   Finally, we highlight three major uncertainties in the tidal response to increasing $CO_2$. The first uncertainty is the increasing $CO_2$ impact on the stratospheric QBO. Although Wang et al. (2022) suggests that tropospheric global
warming increases the frequency of QBO disruption events, the QBO is prescribed with climatology in our future run. Consequently, our simulation does not account for the QBO disruption impacts on the tides, even though such a QBO disruption event has been shown to intensify the (1,1) DW1 tide (Kogure et al, 2021). The second uncertainty is a temperature response in the mesosphere to the $CO_2$ concentration pathway. Garcia et al. (2019) reported that the cooling rate in the lower and middle mesosphere varies depending on the $CO_2$ concentration pathway. While the
RCP 8.5 scenario leads to strong cooling in the lower and middle mesosphere, consistent with our results, this pronounced cooling was absent under the RCP 6.0 scenario. This suggests that the negative tidal trend in the MLT

may vary with the $CO_2$ concentration pathway. The third uncertainty lies in the effects of GWs on the DW1 tide. The positive trends in tropical GW and DW1 tidal source activity are likely robust, as multiple tropospheric climate studies agree that tropical convective activity will increase with rising $CO_2$ levels (e.g., Chou and Neelin, 2004; Feng et al., 2019; Lau et al., 2013). Enhanced tropical convection also leads to increased tidal and GW source activity, as reported by McLandress and Fomichev (2006), Franke et al. (2023), and Watanabe et al. (2005). However, the tidal response to parameterized GW momentum deposition depends on the type of GW parameterization scheme used. Parameterizations based on the Lindzen scheme, used in WACCM-X, dissipate tides, while those based on the Hines scheme intensify tides (McLandress 1997; Mayr et al., 1998; Meyer, 1999). Although GW diffusion always damps tides in both schemes, momentum deposition may mitigate the negative tidal trend. Furthermore, the parameterization used in WACCM-X does not account for horizontal GW propagation (e.g., Sato et al., 2009; Kalisch et al., 2014; Kogure et al., 2018; Song and Chun, 2008; Song et al., 2020) or secondary GW generation (e.g., Becker and Vadas, 2018; 2020; Vadas and Becker, 2018), both of which can significantly influence momentum deposition and diffusion. H. L. Liu (2021, 2025) pointed out that current gravity wave parameterizations, due to their simplifications, significantly underestimate diffusion in the MLT layer. Therefore, a more realistic GW parameterization is necessary to accurately assess the impact of increasing $CO_2$ on the MLT region.

Despite these uncertainties, our study reaffirms that increasing $CO_2$ affects not only the thermal structure, but also the dynamic properties of the MLT region (such as wave activities, diffusion, and circulation) as previously pointed out (Liu et al., 2020). We also confirm the significant negative tidal trend in the MLT from SABER observations over the 23-year period (2002–2024). However, based on the simulation, this 23-year span may be insufficient to fully capture the long-term effects of increasing $CO_2$. Our results indicate that the DW1 (1,1) tidal amplitude decreases over time, although not monotonically, suggesting that the negative tidal trend will be strengthened and robustly confirmed within the next few decades.

**Acknowledgments.**

This research was supported by Global - Learning & Academic research institution for Master's·PhD students, and Postdocs (LAMP) Program of the National Research Foundation of Korea (NRF) grant funded by the Ministry of Education (No. RS-2024-00442483). Huixin Liu acknowledges support by JSPS KAKEN Grants JP25K01058 and JP22K21345.

**Code and Data availability.**

The long-term future simulation in WACCM-X can be downloaded in https://doi.org/10.5281/zenodo.15189573 (Ma et al., 2025). Our python script for Hough mode decomposition can be obtained in the GitHub repository (Kogure ,2025): https://github.com/masaru-kogure/Hough_Function#.

**Competing interests.**

The authors declare that they have no conflict of interest.

**Author contributions.**

MK, IS, and HL conceptualized and designed the study. HLL conducted the long-run simulations. MK analyzed the data and wrote the manuscript draft. All the authors reviewed, edited, and contributed to the scientific discussion in the paper.

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
