# Peer review of "Impacts of Increasing CO2 on Diurnal Migrating Tide in the Equatorial Lower Thermosphere"

_EGUsphere, 2025_

## Author Comment (AC1)

Reply to Reviewer #1.

This paper examines the impact of increasing CO2 on the upward-propagating migrating diurnal tide (DW1) using a WACCM-X simulation with a prescribed long-term surface CO2 trend. The results indicate positive DW1 amplitude responses in the stratosphere and lower mesosphere (20-70 km) and negative responses at higher altitudes (90-110 km). The authors discuss in detail possible mechanisms underlying these altitude-dependent responses.

The study presents new and interesting findings, and I have no fundamental objection to its publication in Atmospheric Chemistry and Physics. However, I see two main weaknesses:

1. The results are not validated against observations, leaving their realism uncertain.

2. The conclusions are drawn from a single simulation without controlled experiments to isolate the suggested mechanisms.

Point (1) could be addressed by referencing existing tidal observations in the literature. Decades of ground-based and satellite measurements are available and should permit at least qualitative comparisons with the simulation results (e.g., consistency in the sign of responses). Point (2) may be more difficult to resolve, given the data storage limitations already noted by the authors. I suggest the authors address at least (1).

Reply: We sincerely thank Dr. Yamazaki for his thoughtful review of our manuscript. We greatly appreciate his comments, which have been invaluable in improving the quality and clarity of our work.

As the reviewer pointed out, it is difficult to resolve Point (2) due to limitations in data storage. We plan to address this in future work. The current paper focuses on Point (1).

To address Point (1), we analyzed temperatures observed by SABER (the Sounding of the Atmosphere using Broadband Emission Radiometry; Russell et al., 1999) instrument aboard NASA's TIMED (Thermosphere Ionosphere Mesosphere Energetics Dynamics) satellite. We added the following discussin in a new Section 5 (lines 359–394 of the revised manuscript).

[revised manuscript text omitted]

The following are other minor comments:

(l. 13) "... the negative trend appears to result from increased eddy diffusion in the mesosphere, ..."
This part is confusing. It first gives a reason for the negative trend, but the following sentence states there are two reasons. I suggest rephrasing for clarity.

Reply: Thank you for the comment. The phrase "to result from increased eddy diffusion" was unintentionally left in the previous manuscript, which may have confused. We apologize for this oversight and have now removed the phase.

(l. 47) "(Yamazaki et al., 2024; Yamazaki and Siddiqui, 2014)"
This should be "(Yamazaki et al., 2014; Yamazaki and Siddiqui, 2024)".

Reply: Thank you for the comment. We have revised it.

(l. 58) "absorption of solar heating"
Perhaps, "absorption of solar radiation"?

Reply: Thank you for the comment. We have revised it.

(l. 96) "and from 2015 onward, it is simulated by rewinding the solar forcing to the 1850 levels (Figure 1b)."
Could you elaborate on this? From 2015 onward, are the f10.7 data theoretical predictions, or are they actual observations from a different time period?

Reply: I appreciate your comment. We used the actual solar observations from 1850, following CMIP6, because, ot he best of our knowledge, it is currently not possible to reliably predict future solar activity.
We have revised the sentences as follows in lines 99–100 of the revised manuscript.
*"... it is simulated using the solar forcing based on historical observational data from 1850 to 1924 levels (Figure 1b). Since future solar forcing cannot be reliably predicted, we used past values following CMIP6."*

(l. 105) "These perturbations were then convolved with the (1,1) Hough mode function to derive the (1,1) mode amplitudes on pressure coordinates."
It should be explicitly stated that the amplitude of the (1,1) Hough mode is computed separately for each pressure surface.

Reply: Thank you for your comment. We have revised the sentence as follows in line 110 of the revised manuscript:
*"These perturbations were then convolved with the (1,1) Hough mode function at each pressure level to derive the (1,1) mode amplitudes."*

(Equation 1)
The right-hand side should be multiplied by Z_{GP}.

Reply: Thank you for pointing out this error. We have added "$z_{GP}$".

(Equation 3)
Should the sign before sigma_i_{2003-2013} be "-" instead of "+"?

Reply: Thank you for pointing out this error. We have corrected the sign in Eqs. (3) and also Eq. (4).

(l. 194) "tidal dumping"
Perhaps, "tidal damping"?

Reply: We have revised it in line 188 of the revised manuscript.

(Figure 4)
I cannot see the ticks on the x-axis. Also, the x-axes look unusual; neither the lower nor the upper x-axis appears to be linearly scaled.

Reply: We have revised Figure 4. Regarding the x-axes, the upper x-axis shows vertical wavenumber on a logarithmic scale, and the lower x-axis shows the corresponding vertical wavelength.
We have revised the caption of Figure 4 as follows:

*"Figure 4. (a) Vertical wavenumbers of DW1 (1,1) mode tides derived from tidal phase values (shown in Figure S2). The upper x-axis denotes vertical wavenumber on a logarithmic scale, and the lower x-axis denotes the corresponding vertical wavelength. The orange and blue lines show vertical wavenumbers averaged over January 2003–December 2013 and December 2050–November 2061, respectively. (b) Difference in vertical wavenumbers between 2003–2013 and 2050–2061. Dashed lines denote the standard errors."*

[Figure]

(l. 223; also in the Figure 5 caption) "Eq. (2)"
Should this be "Eq. (5)"? Eq. (2) does not tell how to calculate the local vertical wavenumber.

Reply: We have revised them in line 230 of the revised manuscript. We appreciate that the reviewer pointed out the errors.

(l. 247) "in zonal mean temperatures (7a) and their vertical gradients (7b)"
Should they be "(6a)" and "(6b)"?

Reply: We have revised them in line 266 of the revised manuscript. We appreciate that the reviewer pointed out the errors.

(l. 256) "Temperatures in this layer are influenced not only by CO2 cooling but also by O3 heating via ultraviolet absorption (Garcia, 2021; Garcia et al., 2019; Jonsson et al., 2004; Lübken et al., 2013)."
Could you describe how O3 changes are produced in the model and how they are connected with the changes in CO2?

Reply: We appreciate your question. We do not claim that changes in CO2 modify O3 chemical production or loss rates, while we believe that changes in CO2 lower the altitude of the ozone layer. We have added the following explanation in lines 284–287 of the revised manuscript to clarify this point .
"This vertical change in $O_3$ concentration is likely due to atmospheric descent caused by the $CO_2$ cooling, as both the positive and negative peaks of the $O_3$ concentration descend by a few kilometers in the future (as shown later in Fig. 8c)."

(l. 308) "In addition,... Therefore, increased CO2 concentrations... and reduce the tidal amplitudes."
I suggest rewriting this part. "Therefore" does not clearly connect the second sentence to the first.

Reply: We appreciate for highlighting the unclear point. We have rewritten the sentence as follows in the lines 343–348.
*"Integrating findings from these previous studies with our simulation, we can lead to the following potential scenario. Increased $CO_2$ concentrations strengthen equatorial convection activity, leading to enhanced tropical gravity wave activity in the stratosphere. These enhanced gravity waves propagate upward and, upon reaching the MLT, saturate and break. This intensifies diffusion and reduces the tidal amplitudes."*

Reply to Reviewer #2.

General Comments:
It is an important topic that the global warming in the stratosphere has possible influences on the dynamics and electrodynamics in middle and upper neutral atmosphere and ionosphere. Tides, which are caused by the solar heating, are an important phenomenon of neutral atmosphere and influence the structures and circulations of middle and upper atmosphere. The migrating diurnal tide with Hough mode of (1, 1) is in the dominant position among the tidal modes in the tropic region. Based on the simulation results of WACCM-X under the scenario of RCP 8.5, this manuscript shows the trends of DW1 (1, 1) from 2000 to 2069. The main results are that the trends of DW1 (1, 1) are the 1% per decade at 20-70 km and -2% per decade at 90-100 km. The possible reasons are the depression in the atmospheric density and enhanced equatorial convective activity and the eddy diffusion.

However, there are some major issues should be fixed. Such as (1) the time period dependency of the linear trend, (1) the reliability of non-LTE CO2 cooling scheme when the concentration of CO2 approaching to its upper limit, (3) the enhanced gravity waves can amplify or depress the tidal amplitude. This is dependent on the relative phases between gravity waves and tides. Here only the depression effect is used. How to exclude the amplification effect.

Please see the detailed comment below.

Reply: We sincerely thank the reviewer for his/her thoughtful assessment of our manuscript. We greatly appreciate the identification of weaknesses, which has been invaluable in improving the clarity and quality of the paper.
(1) Time-period dependency. We agree that the trend seems to vary across analysis periods, although the negative trend in the MLT appears robust. Accordingly, we have removed the quantitative discussion of the trend.
(2) Non-LTE CO2 cooling scheme error. As reported by Fomichev et al. (1998), the error in equator regions is ~1 K/day at 360 ppm and ~2K/day at 720 ppm at maximum around the mesopause, increasing approximately linearly between 150 and 720 ppm. It should be noted that the $CO_2$ concentration at 100 km is ~250 ppm in 2069 and far below 720 ppm; therefore, we consider this error to be negligible.
(3) Gravity wave drag. We agree that diurnal harmonics of gravity wave drag can potentially amplify tides in the real atmosphere. However, the diurnal harmonics of gravity wave drag based on the Lindzen scheme, which WACCM-X used, only dissipate the tide (McLandress 1997; Mayr et al., 1998). Indeed, the tidal amplitudes decrease in the MLT, implying parameterized gravity wave drag dissipates the tide; otherwise, if it amplifies the tide, its positive effect would be negligible. We discussed this uncertainty in Section 6.

Our detailed reply is as follows.

Comments:

1. L10-11: It is better to show the corresponding standard deviations or significant level.
Reply: We thank you for your comment. As the reviewer points out in Comment 6, the magnitude of the negative seems to vary over time. Although the tidal trends are statistically significant, we reconsider that our quantitative estimates might not be meaningless. We therefore have deleted the values of the trend and revised the sentence as follows in lines 9–10 of the revised manuscript.
*"... exhibits a statistically significant positive trend in a range of 20–70 km, and a statistically significant negative trend in a range of 90–110 km."*

2. L79-81: How about the vertical resolution at height above 1 hPa, since the model top is at ~500-700 km, which is much higher than 1 hPa.
Reply: Thank you for the question. The resolution is 0.25 scale height above 1 hPa. We have added the following sentence in line 80 of the revised manuscript for clarification.
*"Above 1 hPa, the resolution is fixed at 0.25 density scale height (e.g., ~1.6, ~1.4, and ~2.6 km at ~70, ~90, and ~110 km altitudes, respectively."*

3. L93: How about the reliability the non-LTE CO2 cooling scheme when the concentration of CO2 is approaching (such as, 600, 650, 700 ppm) to the upper limit (720 ppm).

Reply: Thank you for the insightful question. This scheme is designed for a range of 150–720 ppm (Fomichev et al., 1998), so we regard it as reliable within this range. As reported by Fomichev et al. (1998), the maximum error in the cooling rate in equatorial regions is ~1 K/day at 360 ppm and ~2K/day at 720 ppm around the mesopause. This error seems linearly increasing between 150 and 720 ppm. We added the following sentence in lines 94–96 of the revised manuscript..

*"In equatorial regions, the maximum error in the non-LTE $CO_2$ cooling rate near the mesopause is ~1 K/day at 360 ppm and ~2 K/day at 720 ppm (Fomichev et al., 1998). This error seems linearly increasing from 150 to 720 ppm."*

4. L96-97: "it is simulated by rewinding the solar forcing to the 1850 levels". Please clarify the point of "rewinding".

Reply: Thank you for pointing out the unclarification. We have clarified the point as follows (lines 99–101 of the revised manuscript).

*"... it is simulated using the solar forcing based on historical observational data from 1850 to 1924 levels (Figure 1b). Since future solar forcing cannot be reliably predicted, we used past values following CMIP6."*

5. L123: "It should be noted that the QBO is specified by climatology". I cannot get this statement. Since the QBO's period ranges from 22 to 28 months, and the QBO's amplitude changes from one cycle to another, how about these variations change the trends of DW1? What's the meaning of "specified by climatology"

Reply: Thank you for raising this point. The WACCM-X configuration used in this study has insufficient vertical resolution to internally generate a QBO, and one can be imposed by relaxing the equatorial zonal winds in the stratosphere to the climate-observed QBO zonal winds (H. Liu et al., 2018).

The QBO's period unlikely affects the trend, unless the period drastically changes in the future, because this study focuses on the trend in a few to several decades, which are more than 10 times longer than the QBO's period. Potential uncertainty associated with QBO disruptions is discussed in Section 6.

We revised the sentence in lines 127–130 of the revised manuscript as follows.

*"It should be noted that the WACCM-X configuration used in this study does not internally generate a QBO due to its resolution (H. Liu et al., 2018). Our simulation is imposed by relaxing the equatorial zonal winds to observations for the period of 2000–2015. Beyond 2015, it is imposed to observations from 1959 to 2015.. Because this study focuses on multi-decadal trends, the QBO phase is unlikely to affect the inferred long-term trend."*

6. L125-126: From Figure 2(a, c, d) , it looks like that the amplitude of DW1 (1, 1) mode decreases after ~2028 and more sharper after 2055. This raised the issue of time period dependent linear trend as supposed by Lastovicka & Jelínek (2019, JASTP, Problems in calculating long-term trends in the upper atmosphere, https://doi.org/10.1016/j.jastp.2019.04.01119.04.011). This point should be clarified.

Reply: We appreciate this valuable comment. We agree that the tidal trend seems to vary decade to decade, even though ~30-year windowed trends typically seem negative. Disentangling such temporal dependence from natural interannual variability is challenging, and short-term trends are beyond our scope. To reduce sensitivity to decadal variability and interannual fluctuations (including the 11-year solar cycle), we derive trends using the full 69-year record; this span is ~6 times longer than the solar cycle, which supports the robustness of the negative trend.

We added the following cautionary note in lines 144–149 of the revised manuscript:

*"It should be noted that atmospheric long-term trends can be time-dependent (Lastovicka and Jelínek, 2019). Although our multi-decadal analysis shows statistically significant tidal trends in each layer, except for 70–90 km, this does not imply a monotonic change at the rates listed in Table 2 over short periods (e.g., a single decade). To mitigate the influence of time-dependent variability and interannual fluctuations, we estimate trends using the full 69-year term, which is ~6 times longer than the 11-year solar cycle."*

7. L150-152: This is related to the same issue of the time period dependent of the linear trend.

Reply: Thank you for pointing out the issue. We agree that temporally dependent linear trends may influence the 2003–2013 versus 2050–2061 comparison. Nevertheless, the qualitative outcome is reasonable: the trend of the differences aligns with the layer-wise trends derived from the full~69-year term, even though the magnitudes carry substantial uncertainty. We have added the following sentence in lines 167–169 of the revised manuscript.

*"These differences in the tidal amplitudes are directionally consistent with the linear trends at each layer in the full*

*term (69 years); thus, they provide a qualitative indication of the multi-decadal trend, while the precise magnitudes should be interpreted with caution."*

8. L200-205: From Figure 4b, it looks like that the differences of vertical wavenumbers are not different from zero under 1-sigma in most height range. This indicates that the changes of vertical wavenumbers are not significant. Moreover, the below ~80 km the shorter vertical wavelengths in 2050-2061 contributed to stronger tidal dissipation, why the tidal amplitudes are larger in 2050-2061 (below ~80 km in Figure 3a)

Reply: Thank you for pointing out the point. We agree: the tidal amplitude does not simply follow the changes in its vertical wavenumber. This implies that additional factors (source variability, background density, and GW breaking) also modulate tidal activity (see Section 3). We have revised the sentence in 218–223 lines of the manuscript as follows:

*" However, in 2050–2061 the tidal amplitude is larger within the altitude range of the shorter vertical wavelengths (~54–82 km; compare Fig. 3a and Fig. 4b). This apparent inconsistency indicates that the dissipation due to the decreases in vertical wavelengths is hidden by the enhancement of the tide due to its source activity and reduction of the atmospheric density. It further suggests that, above ~82 km, parameterized gravity wave drag is the primary contributor to the amplitude reduction. The effect of gravity waves will be discussed in Section 4.2."*

9. L257, Equation (4)◊(7), O_{(\delta^2)} should be O(\delta^2)

Reply: Thank you for pointing out the error. We have revised it in line 244 of the revised manuscript.

10. Figures 5-7: It is better to indicate the regions where the significant level is at 1-sigma or 2-sigma.

Reply: We appreciate this comment. We have added the regions in Figs. 5–7 as follows.

[Figure]

Figure 5. (a) Difference in local vertical wavenumber computed from Eq. (5). (b) Effects of buoyancy frequency changes due to increasing $CO_2$ on the local vertical wavenumber. (c) Same as (b), but showing Doppler-shifted effects. Hatched areas indicate regions where the differences are statistically insignificant (i.e., within 1-sigma standard error).

[Figure]

Figure 6. (a) Difference in zonal mean temperature between 2050–2061 and 2003–2013. (b) Same as (a), but showing differences in vertical temperature gradients. Hatched area denotes regions where the differences are statistically insignificant (i.e., within 1-sigma standard error).

[Figure]

Figure 7. (a) Relative difference in $CO_2$ concentration. (b) Same as (a), but for the $O_3$ concentration. Hatched areas indicate regions where the differences are statistically insignificant (i.e., within 1-sigma standard error).

11. L255-281: It is better to convert the concentrations of CO2 and O3 to their corresponding cooling rates, such that one can get their cooling effects more directly.
Reply: Thank you for the valuable comment. We agree with the review. However, the present paper focuses on

mechanisms underlying the negative tidal trend rather than the drivers of strong future mesospheric cooling. A quantitative attribution of mesospheric temperature changes to individual species lies beyond our current scope and will be addressed in a separate study.

We have added the following sentence in lines 307–309 of the revised manuscript.

*"Thus, the vertical changes in the $O_3$ and $CO_2$ concentrations can qualitatively explain the vertical change in the temperature, although a more quantitative evaluation is needed and is beyond the scope of this study."*

[Figure]

12. Figure 8: Please show the error bars or standard deviations in each panel.
Reply: We appreciate this comment. We have added error bars.

Figure 8. (a) Vertical profile of temperature at 1°N in 200301–201312 (blue) and 205012–20611 (orange). (b) same as (a) but for $CO_2$ concentration. (c) same as (a) but for $O_3$ concentration. (d) Difference in temperature at 1°N (205012–20611 minus 200301–201312). (e) same as (d) but for $CO_2$ concentration. (f) same as (d) but for $O_3$ concentration. Dashed lines represent $\pm$1-sigma standard error intervals.

13. L310-312: The enhanced gravity waves may amplify or depress the tidal amplitudes are also dependent on the relative phase between gravity waves and tides (Ortland & Alexander, 2006, JGR, Gravity wave influence on the global structure of the diurnal tide in the mesosphere and lower thermosphere, doi:10.1029/2005JA011467). Why the enhanced gravity waves can only reduce the tidal amplitudes?
Reply: Thank you for the sharp comment. Yes, diurnal harmonics of gravity wave drag can amplify the tide in the real atmosphere. Unfortunately, we do not restore the drag in the high-frequency sampling; thus, we cannot directly evaluate this effect. However, the diurnal harmonics of parameterized gravity wave drag based on the Lindzen scheme, used in WACCM-X, only dissipate tides but do not amplify them. (Meyer, 1999). Indeed, the tide in the

MLT decreases in the future. We have revised the sentence in lines 328–331 of the manuscript as follows. *"However, the gravity wave drag based on the Lindzen scheme, which is implemented in WACCM-X, only acts to dissipate the tide ( Meyer, 1999). Consistent with this, the tidal amplitude in the MLT decreases in the future (see Fig.2c and Fig. 3a)"*

[Figure]

(a) Diff. parameterized GW diffusion
(205012_206111−200301_201312)

(b) Diff. zonal mean presipitation rate
(205012_206111−200301_201312)

14. Figure 9: Please show the confidence level in (a) and standard deviations in (b).
Reply: We appreciate this comment. We have added confidence intervals.

Figure 9. (a) Difference in diffusion due to parameterized gravity waves between 2003–2013 and 2050–2061, shown as a function of latitude and geometric height. Hatched areas indicate regions where the differences are statistically insignificant (i.e., within 1-sigma standard error). (b) Difference in zonal mean precipitation rate between 2003-2013 and 2050-2061. Dashed lines represent $\pm$1-sigma standard error intervals.

15. L329-330: This statement is questionable due to the time period dependent trend. Are the sharp decrease of the tidal amplitude above 70 km caused by the limitation of non-LTE CO2 cooling scheme in the mesosphere due to the CO2 concentration is large in later years.

Reply: We appreciate raising the suspicion. Yes, the scheme has known errors (see Reply to Comment 6) and is validated only up to 720 ppm above ~15 km; therefore, simulations may become unreliable once a $CO_2$ concentration exceeds 720 ppm in the middle atmosphere. We deleted this sentence.

Regarding the sharp decrease in the tidal amplitude above 70 km in ~2055, we believe that this decrease is not attributed to the limitation of the cooling scheme. In ~2055, the $CO_2$ concentrations at 50 km and 100 km remain ~550 and ~210 ppm, respectively (see Figure 1a), which are well below the 720 ppm.

16. L331-334: "We propose two …: a decrease in tidal vertical wavelength and an increase in diffusion due to GW breaking". From Figure 4(a), the vertical wavelengths are longer in 2050-2061 than in 2003-2012 above ~95 km. This is contradictory to "a decrease in tidal vertical wavelength". Why GW breaking can only dissipate tide? It can also amplify tide according to Ortland & Alexander (2006, JGR)

Reply: We thank you for the comments. We agree that the longer vertical wavelengths and gravity wave drag can potentially amplify the tide. However, the longer vertical wavelengths above ~95 km are not significant over most of 95-110 km. Also, the diurnal harmonics of gravity wave drag based on the Lindzen scheme only dissipate the tide (Meyer, 1999). Moreover, if either longer vertical wavelengths or gravity wave drag amplification were dominant there, tidal amplitudes would be expected to increase in the future, which is inconsistent with our main results (Fig. 3a). Therefore, these two amplification mechanisms can be negiligible, though further investigation of gravity wave-tide interactions is warranted. We discuss the uncertainty regarding the gravity wave parameterization in section 6. We have revised the sentences in lines 403–406.

*"We propose two potential mechanisms contributing to the increased tidal dissipation: a decrease in the tidal vertical wavelength and an increase in diffusion due to GW breaking. The decreasing vertical wavelength contributes to the dissipation in ~52–82 km, peaking at ~75 km, while the increasing GW breaking diffusion contributes in ~45–110 km around the equatorial region, peaking at ~93 km."*

---

## Author Response (AR2)

Both reviewers are happy with the changes made to the paper. Please make the following corrections to the revised manuscript:

Reply: We sincerely thank the editor and both reviewers for their thoughtful comments and for taking the time to review the revised manuscript.

Table 1
As the authors have inserted the word "are", the caption now reads:
"Slopes and their standard errors of the linear fit are shown in Figure 2(c-g)."
However, Figure 2 does not show the standard errors. Would not it be more accurate to say something like the following?
"Slopes and standard errors of the linear fits shown in Figure 2(c-g)."
Reply: We removed "are" as suggested (line 158 of the revised manuscript).

Table 2
Although I assume the table presents linear trends in the amplitude of the DW1 (1,1) mode, this is not explicitly stated in either the caption or the main text.

Reply: Yes, it does. We have added "in the amplitude of the DW1 (1,1) mode" in lines 383 and 386 of the revised manuscript.